# Loop Closure Detection for Mobile Robot Based on Multidimensional Image Feature Fusion

**Jinming Li [1], Peng Wang [1,2,\*], Cui Ni [1], Dong Zhang [2] and Weilong Hao [1]**

[1] School of Information Science and Electrical Engineering, Shandong Jiaotong University, Jinan 250357, China
[2] Institute of Automation, Shandong Academy of Sciences, Jinan 250013, China
\* Correspondence: 205049@sdjtu.edu; Tel.: +86-15753177015

**Abstract:** Loop closure detection is a crucial part of VSLAM. However, the traditional loop closure detection algorithms are difficult to adapt to complex and changeable scenes. In this paper, we fuse Gist features, semantic features and appearance features of the image to detect the loop closures quickly and accurately. Firstly, we take advantage of the fast extraction speed of the Gist feature by using it to screen the loop closure candidate frames. Then, the current frame and the candidate frame are semantically segmented to obtain the mask blocks of various types of objects, and the semantic nodes are constructed to calculate the semantic similarity between them. Next, the appearance similarity between the images is calculated according to the shape of the mask blocks. Finally, based on Gist similarity, semantic similarity and appearance similarity, the image similarity calculation model can be built as the basis for loop closure detection. Experiments are carried out on both public and self-filmed datasets. The results show that our proposed algorithm can detect the loop closure in the scene quickly and accurately when the illumination, viewpoint and object change.

**Keywords:** computer vision; visual SLAM; loop closure detection; feature fusion

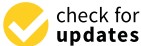



## 1. Introduction

Vision-based simultaneous localization and map building (VSLAM) refers to a mobile robot outfitted with vision sensors that acquire and analyze images of its surroundings in order to position itself and build a map in real-time. Currently, VSLAM is widely used in robotics and visual navigation. During the operation of VSLAM, both the visual sensor and the robot's motion estimation will generate errors, and if the errors continue to accumulate, they will seriously impair the mobile robot's judgment of its positioning and may even result in map construction failure [1]. Loop closure detection [2] is primarily used to determine whether the mobile robot's current position has been visited previously, which can effectively reduce the accumulated errors and achieve repositioning after the robot has lost its position, helping to ensure accurate positioning and map construction. The loop closure detection in the existing VSLAM system primarily determines whether the loop closure can be formed by calculating the similarity between the current frame and the keyframe; thus, loop closure detection is fundamentally a scene recognition and image matching problem [3,4], which compares the current scene in which the mobile robot is located in with the historical scene to determine whether the scene is the same. In real-world scenes, however, changes in location, illumination, viewpoint and object can cause changes in the visual appearance of the environment, affecting the accuracy of loop closure detection and increasing the difficulty of detection. Currently, this is a pressing issue in loop closure detection.

Traditional loop closure detection methods rely on human-designed feature descriptors [5] and are primarily implemented based on appearance. There are two types of human-designed feature descriptors: local descriptors and global descriptors [6]. Local descriptors (such as SURF [7], SIFT [8], BRIEF [9] and others) can extract various detailed

features from images quickly and effectively, and have advantages such as scale and rotation invariance. However, local descriptors focus on the image's local features, cannot fully characterize the entire image and perform poorly when dealing with illumination and object changes. Global descriptors (such as Gist [10], HOG [11], color histogram [12] and others) can be obtained by compressing and abstracting the entire image; they can better describe the image as a whole and perform better when the environment changes. For example, in [13], Qiu et al. took Gist features as the target of a convolutional self-coding network reconstruction, which could enhance the expression ability of scene features of the model through changes in appearance. However, because it completely abandons the image's detailed features, the accuracy is significantly reduced when the image viewpoint and illumination change significantly. As can be seen, the above feature descriptors can only obtain shallow information about the image, posing significant limitations and making them difficult to widely apply in environments with complex scene content and large changes, which limits the applicability of loop closure detection. As a result, the traditional loop closure detection method is incapable of meeting the accuracy requirements of VSLAM.

With the advancement of deep learning, some researchers have begun to focus on deep learning in the hope that deep neural networks can be used to improve the performance of loop closure detection and solve problems that traditional loop closure detection methods cannot handle, and significant research results have been obtained. The deep learning-based loop closure detection method involves feeding the acquired environmental images into a deep neural network for processing and feature extraction, and obtaining a variety of information from the images, such as semantic, color, geometry, etc.; in this way, multiple features can be used to describe the images in different dimensions, which can be used to more comprehensively and accurately determine the similarity between images [14]. Furthermore, the deep learning-based loop closure detection method has a broader range of applicability and stronger generalization ability, and can cope with changes in illumination and viewpoints in the scene with greater robustness. Gao et al. [15] and Liu et al. [16] used a deep neural network to extract image features for loop closure detection, which significantly improved the accuracy compared with traditional methods. In [17], Sünderhauf et al. used the AlexNet network to evaluate the effect of scene changes on loop closure detection, and discovered that the third convolutional layer is optimal for scene recognition, but the dimensionality of its output feature vector is too large to meet the real-time loop closure detection requirements. Deep learning techniques are now widely used in image recognition and feature extraction, with impressive results in image classification, target detection, semantic segmentation and other fields. Cao et al. [18] used object detection to obtain feature nodes and construct subgraphs. After comparing subgraphs, two similarity matrices were obtained and global similarity scores were calculated, which were used as the basis for loop closure detection. This method is less sensitive to illumination changes and more adaptable to application scenarios. Garg et al. [19] and Hausler et al. [20] used deep neural networks to build a visual position recognition system, which they used to generate and integrate a variety of descriptors for robot position recognition, and were able to achieve ideal results.

As a typical application of deep learning in vision, semantic segmentation can accurately segment objects in images and obtain more image information to accurately characterize objects based on segmentation boundaries. Semantic segmentation has gradually been applied in the field of loop closure detection in recent years. For example, in [21], Li et al. used semantic information to exclude dynamic target interference for loop closure detection, obtaining high- and low-dimensional convolutional neural network (CNN) features of images, and combining CNN features of different dimensions for loop closure detection. In [22], Wu et al. applied semantic segmentation to extract semantic information of images before calibrating the convolutional features acquired by a CNN using the semantic features, constructed the descriptors (TNNLoST), and finally, completed the loop closure detection based on the TNNLoST. This method combined semantic and convolutional features to improve loop closure detection. To extract semantic labels and

combine visual information, Yuan et al. also used semantic segmentation in [23]. They created a semantic landmark vector model and fused and calculated multiple feature information to complete the closed-loop judgment. Although the semantic segmentation-based loop closure detection methods have a higher accuracy and better performance than the traditional loop closure detection methods, they still have many shortcomings, such as the long training and running time of the models, noise easily occurring in the process of semantic segmentation, difficulty adjusting parameters in the network, and difficulty realizing the real-time operation of the loop closure detection algorithm due to the device's high computing power requirement, which is a very challenging problem nowadays.

In order to solve the shortcomings of the traditional loop closure detection methods and the deep learning-based loop closure detection methods, we propose a loop closure detection algorithm that fuses Gist, semantic and appearance features; the algorithm process is depicted in Figure 1. Firstly, the Gist feature vectors in the current frame and all keyframes are extracted and matched, and the keyframes with the highest Gist similarity to the current frame are chosen as loop closure candidates, with each candidate frame assigned a Gist similarity score. The current and candidate frames are then semantically segmented, where each mask block's area proportion in the segmentation result is computed; fine mask blocks with too small a proportion are discarded, and the area proportion is used as a weight in the calculation of the local similarity score. Following that, the semantic labels output the segmentation results one by one. After obtaining the center of mass for each mask block in the segmentation results, the three-dimensional (3-D) semantic nodes can be built by combining the image depth information, and the cosine similarity of the semantic nodes in the two frames can be compared one by one to obtain the semantic similarity score of each node pair, where the node pair with a higher similarity is chosen as the similar node pair. The shape similarity of the two mask blocks in the similar node pairs is then compared one by one to determine the appearance similarity score, and the local similarity score of the image is calculated after weighted fusion with the semantic similarity score. Finally, by combining the Gist similarity score and local similarity score, the final similarity score of the image can be obtained, and the loop closure detection will be completed according to this score.

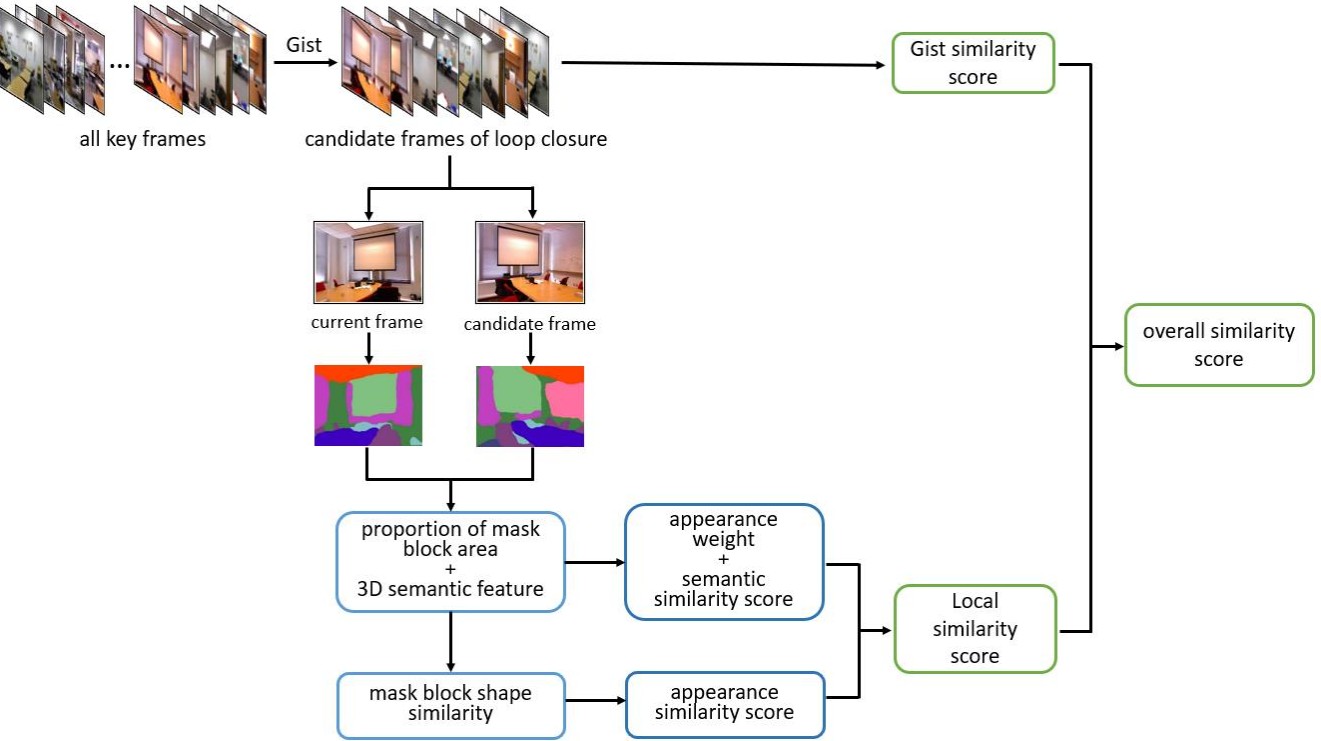

**Figure 1.** Flowchart of the proposed algorithm.

In summary, the following are the study's main contributions:

1.  A multidimensional image similarity comparison model incorporating Gist features, semantic features and appearance features is built for detecting loop closures of mobile robots in indoor environments;
2.  A multilayer screening mechanism is used to quickly screen loop closure candidate frames using Gist features to save computational resources, and the constructed multidimensional image similarity comparison model is used to accurately extract real loop closures from candidate frames to improve loop closure detection accuracy.

The rest of this paper is organized as follows: Section 2 briefly discusses the relevant work of the method in this paper, Section 3 describes the specific method in this paper, and Section 4 is the detailed analysis and comparison of the experimental results. Finally, the findings of this paper are summarized in Section 5.

## 2. Related Work

### 2.1. Gist Feature

Gist is a bio-inspired feature that describes the global features of a scene based on the spatial envelope model of the scene image, such as roughness, naturalness, openness, dilation and precipitousness, and can be used to obtain local structure information of the scene image at different spatial frequencies, locations and orientations via a Gabor filter set, which has achieved good results in image scene-recognition classification tasks. Gist features are used to quantify the scene distribution features of an image in order to describe the image's scene distribution content. The image does not need to be segmented or local features extracted. This feature, when compared to local features, is a more "macroscopic" feature description method that can be used in closed-loop detection.

When Gist features are extracted, the image is first divided into multiple subregions, as shown in Figure 2, and then, the Gabor filter bank is used to perform convolution filtering on each subregion to extract Gist features of each, as shown in Equation (1):

$$Gist_i(x,y) = cat[f^i(x,y) * g_{mn}(x,y)] \tag{1}$$

where $cat$ denotes the cascade operation of a subregion after the Gabor filter set, $f^i(x,y)$ denotes the $i$-th subregion of the image, $*$ denotes the convolution operation and $g_{mn}(x,y)$ denotes the two-dimensional Gabor function. Finally, the average value of the Gist features of each subregion is calculated, and the Gist features of the entire image are obtained by cascading the average value of the Gist features of each subregion.

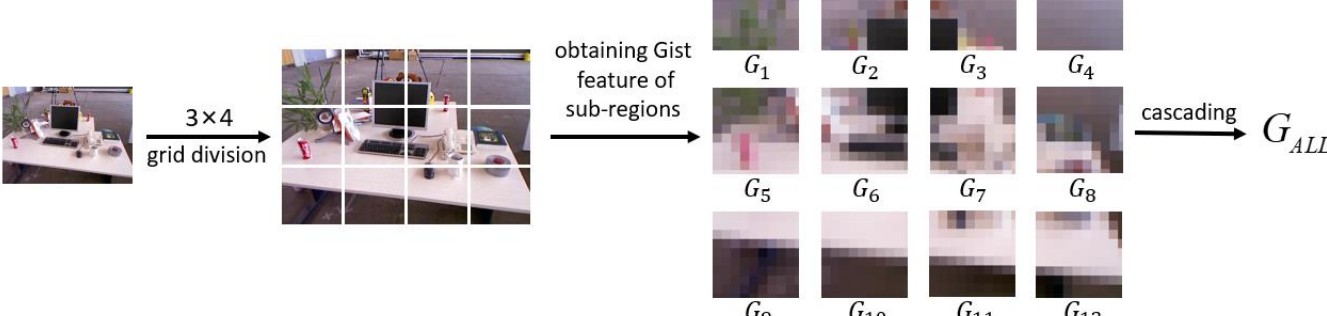

**Figure 2.** Flowchart of global Gist feature extraction (taking $3 \times 4$ grid division as an example).

### 2.2. Semantic Segmentation Network

Semantic segmentation is a key component of the computer vision field. Semantic segmentation combines image classification, target detection and image segmentation through the classification of the image pixel by pixel, segmentation of the image into regional blocks with a specific semantic meaning and identification of the semantic category

of each regional block, completing the semantic reasoning process from bottom to top, and finally, producing a segmented image with pixel-by-pixel semantic annotation. Semantic segmentation is less sensitive to changes in illumination, object morphology, etc., and can segment the current environment more precisely, resulting in more accurate image information. U-Net [24], PSPNet [25] and DeepLab series algorithms [26–28], among others, are common semantic segmentation algorithms used today. DeepLab series algorithms have the best segmentation accuracy and segmentation effect. As shown in Figure 3, the original Deeplabv3+ model employs Xception [29] as the backbone network and employs depth-separated convolution in Xception to improve Deeplabv3+ segmentation and the segmentation effect. However, the number of parameters in Xception is excessive, resulting in a slow running speed. The network structure of mobileNetv2 [30], a lightweight network proposed for small embedded devices, is shown in Figure 4. MobileNetv2 improves both training and running speed while maintaining accuracy; thus, we improve the Deeplabv3+ semantic segmentation model by using MobileNetv2 as the backbone network, which achieves the goal of shortening the model's training and running time and compensating for the disadvantage of indoor mobile robots' limited arithmetic power. We use the Deeplabv3+ semantic segmentation network in this paper to segment keyframes, obtain image segmentation results, divide various objects in keyframes based on semantic labels and assign mask blocks with corresponding colors. The mask blocks and depth information are used to build 3-D semantic nodes, and the semantic nodes are used to characterize the semantic features, which are then used to calculate the semantic similarity score and the final image similarity comparison model.

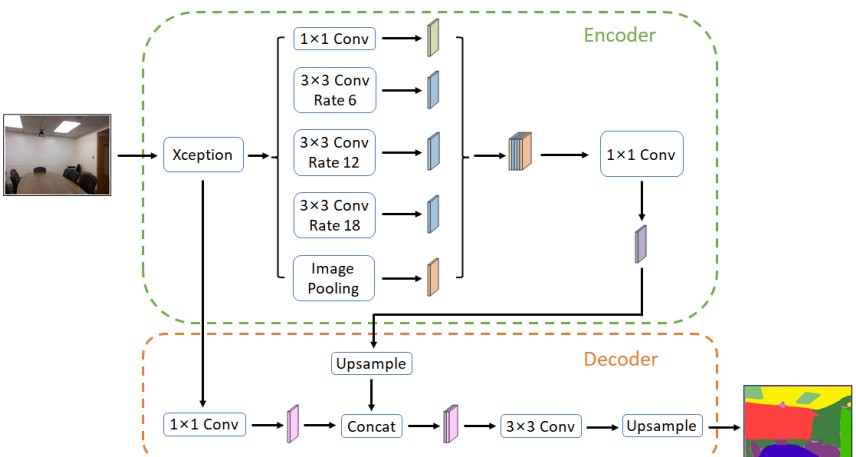

**Figure 3.** Deeplabv3+ network structure diagram.

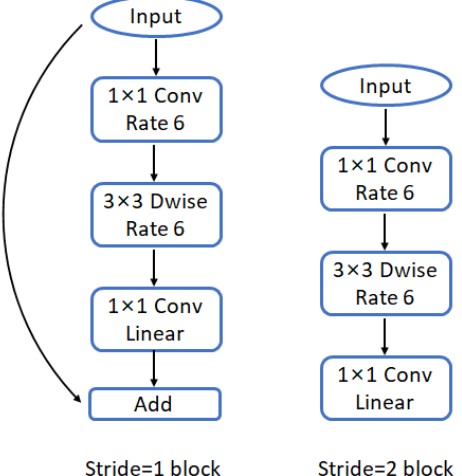

**Figure 4.** Structure diagram of MobileNetv2.

*2.3. Moment*

Moments are the weighted average of some specific pixel grayscales of an image, or a property of an image with a similar function or meaning, which are used to characterize the geometric representation of an image region, and are also known as geometric moments in the field of computer vision, digital image processing, etc. Geometric moments of various orders are used to represent various physical properties of an image, such as zero-order moments for computing a figure's area, first-order moments for computing a figure's center of mass, etc. Image retrieval and recognition, image matching, image reconstruction, digital compression, digital watermarking and motion-image sequence analysis all use geometric moments. The algorithm in this paper calculates the zero-order moment of each mask block to obtain the area of the mask block in the image, and the zero-order moment is calculated as shown in Equation (2). The first-order moment is used to calculate the centroid coordinate of each mask block, where it is regarded as a two-dimensional (2-D) semantic node, and the first-order moment calculation is shown in Equation (3).

$$m_{00} = \iint f(x,y)dxdy \tag{2}$$

$$\begin{aligned} m_{10} &= \sum_x \sum_y x f(x,y) \\ m_{01} &= \sum_x \sum_y y f(x,y) \end{aligned} \tag{3}$$

where $f(x,y)$ denotes a 2-D grayscale image.

Hu moments [31], a representative class of geometric moments composed of seven invariant moments, can more accurately describe the shape of a figure and are frequently used to calculate the shape similarity between figures. Additionally, Hu moments have translation, scaling and rotation invariance, allowing them to better deal with the translation and rotation that occurs during the acquisition of environmental images [32]. Hu moments are fast to calculate but have a low recognition accuracy; they are frequently used in images to identify objects with larger areas or clearer contours. As a result, we use the Hu moment to calculate the shape similarity of large mask blocks, except of noise, in order to extract the appearance features of each object in the image quickly and accurately, which can then be used in the subsequent calculation of the appearance similarity score and the final image similarity comparison model.

## 3. Methodology

*3.1. Screening of Loop Closure Candidate Frames*

Mobile robots continuously acquire peripheral images during VSLAM navigation, which are prone to problems such as loop closure detection miscalculation and computational redundancy due to the high repetition of these images. The ideal loop closure detection algorithm must not only be accurate, but it must also be able to operate in real-time. The current deep learning-based loop closure detection algorithm is significantly more accurate than the traditional loop closure detection algorithm, but it is slow, has a large number of redundant calculations and has other issues. The traditional loop closure detection algorithm runs quickly, but is prone to false-positive loop closure judgments. To improve the algorithm's execution efficiency while ensuring loop closure detection accuracy, this paper uses Gist global features to initially screen all keyframes, where keyframes with a higher Gist similarity are used as loop closure candidates that participate in the subsequent loop closure detection steps; this process not only reduces the number of images processed by the subsequent semantic segmentation model and saves a lot of unnecessary computing power and time by processing images with too little similarity, but it also ensures the algorithm's accuracy.

During the operation of the mobile robot, a large number of redundant images with a high similarity will be collected. These redundant images will waste a significant amount of time and computing resources to be processed. As a result, a screening strategy for extract-

ing keyframes from a large number of historical images and using keyframes to represent local continuous image sequences is required to properly reduce image redundancy and ensure the system's smooth operation. In this paper, the time-interval constraint method is used to extract keyframes, which means that a keyframe is extracted at each interval in a continuous image.

Assume that the current frame is $f_c(x_c, y_c)$, the keyframe set is $F = \{f_1(x_1, y_1), f_2(x_2, y_2), \cdots, f_n(x_n, y_n)\}$, and the resolution of $f_c(x_c, y_c)$ and all keyframes contained in the $F$ sets are $W \times H$. A keyframe $f_k(x_k, y_k)$ is obtained from $F$, and then there is a grid division of $f_c(x_c, y_c)$ and $f_k(x_k, y_k)$ with R rows and C columns; next, $R \times C$ subregions are obtained with a number for each subregion of $1, 2, 3, \cdots, (R \times C)$. Then, $f_c(x_c, y_c) = \{f_c^{\ 1}(x_c, y_c), f_c^{\ 2}(x_c, y_c), \cdots, f_c^{\ I}(x_c, y_c)\}$ and $f_k(x_k, y_k) = \{f_k^{\ 1}(x_k, y_k), f_k^{\ 2}(x_k, y_k), \cdots, f_k^{\ I}(x_k, y_k)\}$, in which each subregion size is $w \times h$, where $w = W/R$ and $h = H/C$. A Gabor filter set is constructed for the m-channel and n-direction of the single-channel image, convolution filtering is performed on the $R \times C$ subregions of $f_c(x_c, y_c)$ and $f_k(x_k, y_k)$ and the local Gist features of each subregion are obtained according to Equation (1). The Gist feature vector of each subregion is obtained by taking the average of the local Gist features of each subregion as $G_1, G_2, G_3, \cdots, G_I$, and cascading the Gist feature vectors of each subregion to construct the global Gist feature vector $G_{ALL} = [G_1, G_2, G_3, \cdots, G_I]$. The global Gist feature vectors of $f_c(x_c, y_c)$ and $f_k(x_k, y_k)$ are $G_{ALL}^c$ and $G_{ALL}^k$; $G_{ALL}^c$ and $G_{ALL}^k$ can be compared to obtain the Gist similarity score *Gscore* between the current frame $f_c(x_c, y_c)$ and the keyframe $f_k(x_k, y_k)$. The *Gscore* of all keyframes in $F$ can be calculated, the Gist similarity score's threshold of $\theta_g$ can be set, and the keyframes with a higher score than $\theta_g$ can be filtered out and as the loop closure candidate frames.

By filtering the loop closure candidate frames, a large number of keyframes with a low similarity can be filtered out to avoid wasting time and arithmetic power on low-similarity keyframes, to shorten the running time of the algorithm and to help to improve the real-time performance of the loop closure detection algorithm.

### 3.2. Calculation of Local Similarity

Assuming that the loop closure candidate frame set is $F_{can} = \{f_{can}^1, f_{can}^2, \cdots, f_{can}^m\}$, Deeplabv3+ can be used for semantic segmentation of the current frame $f_C$ and all loop closure candidate frames in $F_{can}$, where the semantic segmentation result of $f_C$ is $s_C$, and the semantic segmentation result set of $F_{can}$ is $S_{can} = \{s_{can}^1, s_{can}^2, \cdots, s_{can}^m\}$. The schematic diagram of the semantic segmentation result is shown in Figure 5.

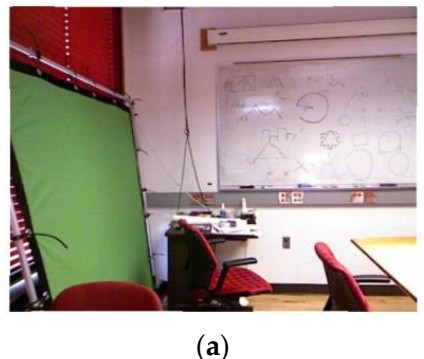 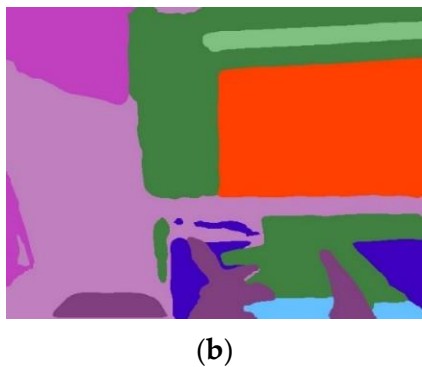

           **(a)**                                        **(b)**

**Figure 5.** Schematic diagram of the semantic segmentation results. (**a**) is the RGB image, and (**b**) is the semantic segmentation result of the original image. Each color mask block represents a semantic label, i.e., each object is given a color; for example, the red mask block in (**b**) represents the label "whiteboard" and the dark green mask block represents the label "wall".

### 3.2.1. Mask-Block Weight

The current frame is semantically segmented with the loop closure candidate frame, and the mask blocks are sequentially outputted by semantic labels to separate the mask blocks from the image after the semantic segmentation result is obtained, as shown in Figure 6. Due to the inherent flaws of the semantic segmentation model, a significant amount of noise interference occurs frequently during the semantic segmentation process, and these noises are typically fine, densely distributed and irregular. It not only wastes a lot of time and arithmetic power to deal with these noises in the subsequent semantic node construction and matching process, but it also easily leads to feature mismatching due to the dense noise distribution. To solve the above problem, we used geometric moments to calculate the area of the mask block in the segmentation result, discarded the mask block with a too small area, calculated the proportion of the area of the mask block in the image and assigned the proportion as a weight to the mask block. If the mask block's area is less than the specified threshold $\theta_a$, it is considered noise and discarded, and it is not used in the subsequent area-proportion calculation.

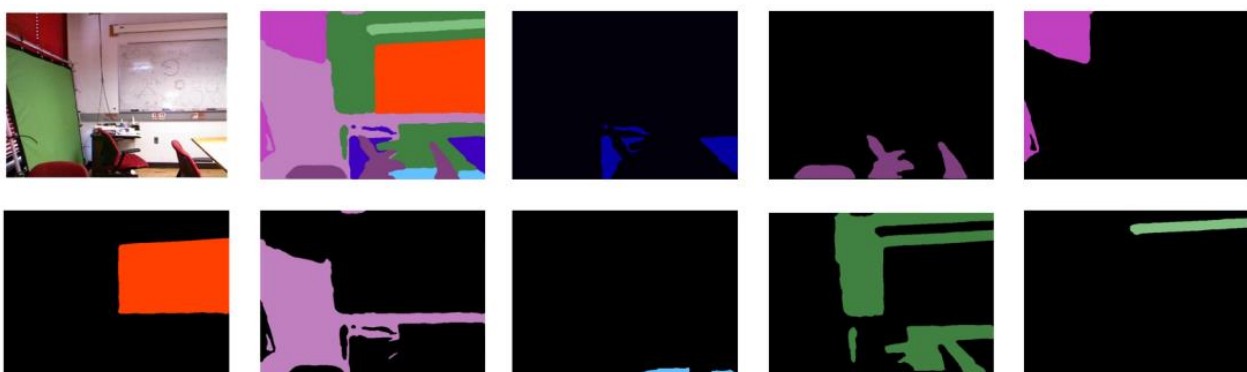

**Figure 6.** Masks are classified by semantic labels. The original RGB image is the first image on the top left, the result of semantic segmentation is the second; the other images are mask images based on the semantic labels.

In general, the greater the proportion of the object's area in the image, the more important it is for determining image similarity and the higher the level of confidence in the subsequent matching process. As a result, we use the area proportion as the mask block's appearance weight and contribute to the calculation of the local similarity score. Following area-based noise filtering, the total area of the remaining mask blocks is calculated, and the area proportion of each mask block is calculated based on the total area and used as the appearance weight $w$:

$$w_l^i = \frac{m_l^i}{W \times H - N} \tag{4}$$

where $w_l^i$ denotes the appearance weight of the $i$-th mask block contained in the $l$th class of semantic labels in the semantic segmentation result, and its value range is [0, 1]; $m_l^i$ denotes the area of this mask block; $W$ denotes the width of the image; $H$ denotes the height of the image; and $N$ denotes the sum of the area occupied by all the noise.

### 3.2.2. Semantic Similarity Score Calculation

Following the methods and calculations in Section 3.2.1, all of the remaining mask blocks in the semantic segmentation results are assumed to have actual meanings, where each mask block corresponds to an object that is actually present in the indoor environment image. If the two frames are similar, the semantic segmentation results must include a large number of semantic labels that are identical. This paper compares the types of semantic labels contained in the current semantic segmentation results to further filter out the less similar candidate frames. As shown in Figure 7, firstly, all of the semantic labels from the

results of the current frame segmentation $s_c$ and a candidate frame segmentation result $s_{can}$ are extracted and expressed as $sl_c$ and $sl_{can}$, and the total number of semantic labels that are the same across the two frames is $sl_{same}$. Then, $\theta_l$ is set to filter through all the candidate frames, and if Equation (5) is satisfied, the candidate frame is retained; otherwise, the candidate frame is removed from the loop closure candidate frame set.

$$\begin{cases} sl_{same} > \theta_l \times sl_c \ , \ sl_c \leq sl_{can} \\ sl_{same} > \theta_l \times sl_{can} \ , \ sl_c > sl_{can} \end{cases} \tag{5}$$

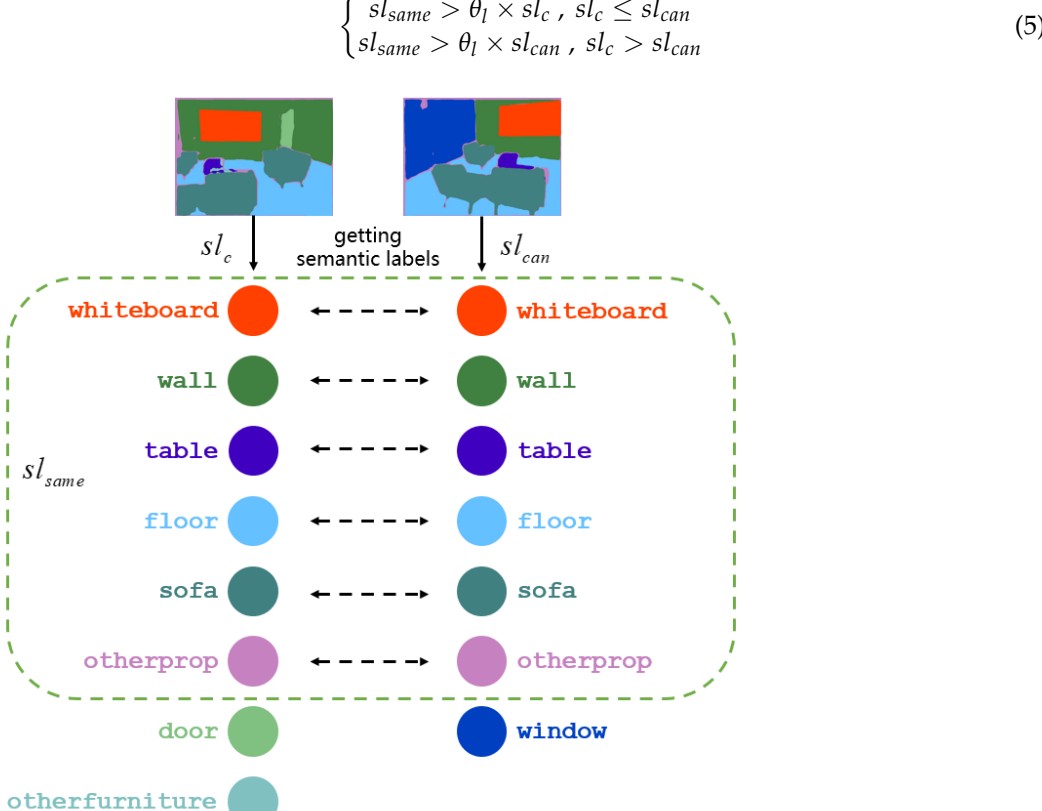

**Figure 7.** Semantic label acquisition and comparison.

The 3-D semantic nodes of the current frame and the filtered loop closure candidate frames are built separately. After noise removal, the semantic segmentation results of the two frames are obtained. The semantic segmentation results contain various semantic labels, and in accordance with the semantic labels, the matching mask blocks are output one by one. Next, the mask blocks contained in each semantic label are extracted one by one to obtain their centroid $C = (X, Y)$, and the centroid $C$ is taken as the 2-D semantic node of the current mask block, as shown in Figure 8. When the indoor mobile robot acquires the environment image using the depth camera, the RGB image of the environment and the corresponding depth image can be acquired at the same time. Then, according to the coordinates of $C$, the depth of the pixel at the appropriate place is acquired from the depth image that corresponds to the RGB image, and it is used as the depth information $Z$ to construct the 3-D semantic node $C = (X, Y, Z)$ of the current mask block. Finally, the 3-D semantic nodes of all mask blocks contained in one semantic label are output in turn, and each semantic node is saved via semantic label classification. Each object in the environment image can be accurately characterized using the 3-D semantic nodes.

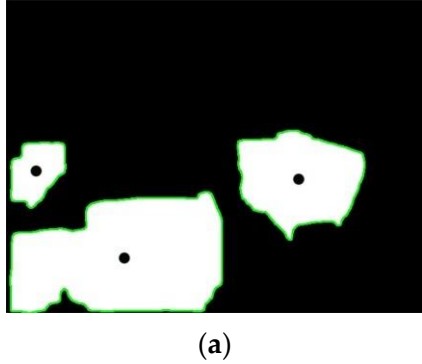 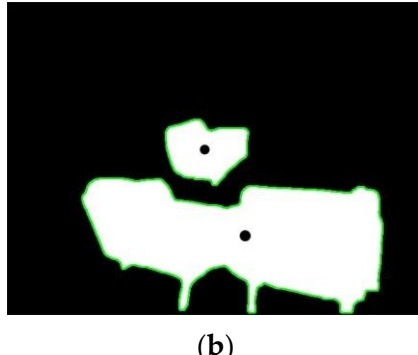

(**a**)    (**b**)

**Figure 8.** Extraction of 2-D semantic nodes. (**a**) shows the 2-D semantic nodes contained in the "sofa" label in the current frame; (**b**) shows the 2-D semantic nodes contained in the "sofa" label in the candidate frame.

The cosine similarity of the semantic nodes in the two frames is compared after obtaining all of the 3-D semantic nodes of the current frame and the candidate frame. Semantic label classification extracts all 3-D semantic nodes in the current and candidate frames that belong to the semantic label. Assuming that the set of semantic nodes belonging to the semantic label l in the current frame is $S_{cl} = \left\{ C_{cl}^1, C_{cl}^2, \cdots, C_{cl}^k \right\}$, and the set of semantic nodes belonging to the semantic label l in the candidate frame is $S_{canl} = \left\{ C_{canl}^1, C_{canl}^2, \cdots, C_{canl}^m \right\}$, then a 3-D spatial coordinate system is established, with the upper left corner of the image serving as the coordinate origin, and the upper, left and depth of the image as the axes. $C_{cl}^i = (X_1, Y_1, Z_1)$ and $C_{canl}^j = (X_2, Y_2, Z_2)$, and then the cosine calculation formula of the corresponding semantic nodes in the current frame and the candidate frame can be expressed as Equation (6):

$$\cos(C_{cl}^i, C_{canl}^j) = \frac{X_1 X_2 + Y_1 Y_2 + Z_1 Z_2}{\sqrt{X_1^2 + Y_1^2 + Z_1^2} \times \sqrt{X_2^2 + Y_2^2 + Z_2^2}} \tag{6}$$

where $C_{cl}^i$ denotes the *i*-th semantic node belonging to semantic label l in the current frame, $C_{canl}^j$ denotes the *j*-th semantic node belonging to semantic label l in the candidate frame and $\cos(C_{cl}^i, C_{canl}^j)$ denotes the cosine similarity of the node pair formed by $C_{cl}^i$ and $C_{canl}^j$. Every semantic node in sets $S_{cl}$ and $S_{canl}$ that belongs to semantic label l is compared individually for cosine similarity in the current frame and candidate frame. In this way, the cosine similarity of all nodes belonging to the same semantic label in two frames can be compared one by one. The semantic similarity score of the node pairs is used in the subsequent calculation of the local similarity score.

Setting the cosine similarity threshold $\theta_c$, the node pairs with a similarity greater than $\theta_c$ can be taken as similar node pairs and their number can be counted If the ratio of the number of similar node pairs to the number of all node pairs is less than $\theta_s$, it can be considered that there is no loop closure between the current frame and the candidate frame; thus, the candidate frame can be discarded, and the comparison with the next candidate frame can occur. If the ratio of the number of similar node pairs is greater than $\theta_s$, it is decided that the two frames may constitute a loop closure, and the comparison is continued in the subsequent steps. After this screening, the comparison range for subsequent loop closure detection can be further reduced, saving time and arithmetic power.

3.2.3. Appearance Similarity Score Calculation

We use Hu moments in this paper to calculate the shape similarity of every mask block in the current and candidate frames. After the noise is removed, the remaining individual mask blocks occupy a larger area in the image and have a clearer contour in the semantic

segmentation result, allowing for an accurate calculation of the shape similarity between the two frames using Hu moments.

The similar node pairs belonging to semantic label l in the semantic segmentation results $s_c$ and $s_{can}$ are extracted, the shape similarity of the two mask blocks contained in the similar node pairs are compared one by one and the results are calculated, as shown in Figure 9. In order to facilitate the subsequent calculation of the local similarity score, this paper uses Equation (7) to normalize the calculation results of the Hu moments and control its value range between [0, 1].

$$Sh_l^i = \frac{2}{\pi} \times \arctan |hu_l^i| \tag{7}$$

where $Sh_l^i$ denotes the appearance similarity score of the *i*-th similar node pair belonging to semantic label *l* in $s_c$ and $s_{can}$, and $hu_l^i$ denotes the shape similarity of the two mask blocks in this node pair.

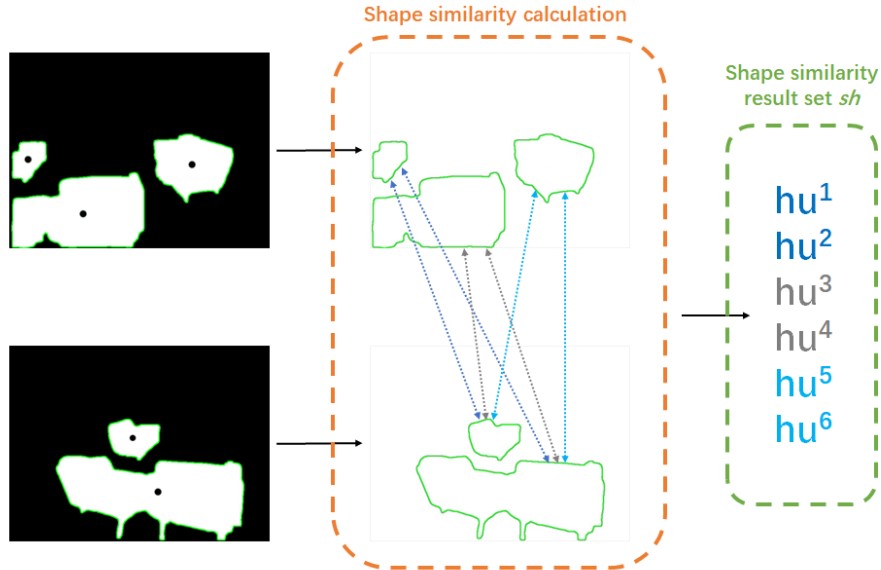

**Figure 9.** Calculation of shape similarity. According to the contour of mask blocks, Hu moment is used to calculate the shape similarity between mask blocks one by one, and the calculation result sh is obtained.

Assuming that the number of similar node pairs belonging to label l in $s_c$ and $s_{can}$ is *p*, the set of the shape similarity calculation results is $sh_l = \left\{ hu_l^1, hu_l^2, \cdots, hu_l^p \right\}$. When using Equation (7) to calculate the shape similarity of mask blocks, the closer the calculation result is to 0, the higher the shape similarity of the two mask blocks; conversely, the closer the calculation result is to 1, the lower the shape similarity of the mask blocks.

### 3.2.4. Local Similarity Score Calculation

In this section, the semantic similarity scores computed in Section 3.2.2 and the appearance similarity scores computed in Section 3.2.3 are weighted and fused to obtain the local similarity scores between each similar node pair. Assuming that the *i*-th feature belonging to semantic label l in the current frame is $M_{cl}^i$ and the *j*-th feature belonging to semantic label l in the candidate frame is $M_{canl}^j$, the local similarity of the node pair can be obtained by using Equation (8).

$$Pscore(M_{cl}^i, M_{canl}^j) = ave(w_{cl}^i, w_{canl}^j) \cdot Se_l^{ij} \cdot (1 - Sh_l^{ij}) \tag{8}$$

where *Pscore* denotes the local similarity score of the similar node pair, $w_{cl}^i$ and $w_{canl}^j$ are the appearance weights of the two mask blocks in the node pair, $Se_l^{ij}$ denotes the semantic similarity score of the similar node pair and $Sh_l^{ij}$ denotes the appearance similarity score of the similar node pair.

Finally, the local similarity score of image can be obtained by accumulating the calculated local similarity of each similar node pair. Assuming that there are n similar node pairs, it the calculations are carried out using Equation (9):

$$Lscore = \sum_1^n Pscore \qquad (9)$$

where *Lscore* denotes the local similarity score between the current frame and the candidate frame, and *Pscore* denotes the similarity score of each node pair within the above two frames.

*3.3. Final Similarity Calculation*

After the calculations from Sections 3.1 and 3.2 are carried out, the Gist similarity score *Gscore* and the local similarity score *Lscore* between the current frame and the loop closure candidate frame can be obtained, respectively, where *Gscore* is calculated by compressing and abstracting the whole image using Gist descriptors to describe the overall features of the environment image, and *Lscore* is calculated by combining the semantic features and appearance features of the image, which is more inclined to describe the local features of the image. In order to describe the whole image more accurately, we adopted Equation (10) to construct an image similarity comparison model by using the weighted fusion of *Gscore*, which represents the image global features, and *Lscore*, which represents the local features, to calculate the final similarity score *Fscore* of the image and to characterize the whole image content as comprehensively and accurately as possible.

$$Fscore = \alpha \cdot \tan(\frac{\pi}{4} \cdot Gscore) + (1 - \alpha) \cdot \tan(\frac{\pi}{4} \cdot Lscore) \qquad (10)$$

where $\alpha$ is the weighting coefficient. Since the calculation process in Section 3.2 is based on the loop closure candidate frames screened in Section 3.1, the *Gscore* values do not differ much and are generally high when calculating the final similarity of the images, whereas *Lscore* can more obviously show the difference between the two frames. Therefore, in this paper, $\alpha$ is set to 0.3 when calculating *Fscore*, and the similarity between images is more accurately represented by adjusting the weight of *Gscore* and *Lscore*.

**4. Experiment and Analysis**

In order to verify the performance of our proposed loop closure detection algorithm, the performance of the Deeplabv3+ semantic segmentation model equipped with various backbone networks is first compared, and the optimized Deeplabv3+ model is pretrained using the NYUv2 dataset after a suitable backbone network is selected. The proposed algorithm is then verified by conducting experiments on public datasets, including the TUM RGB-D dataset, the Microsoft 7-Scenes dataset, the RGBD Scenes dataset v2 and a self-filmed dataset, and it is compared in detail with the DBoW [33], CNN-W [16], algorithm from reference [18] (Cao.) and algorithm from reference [22] (Wu.).

*4.1. Datasets*

The TUM RGB-D dataset [34] is a public dataset, which was created from images taken of various scenes using a Microsoft Kinect camera and which contains a total of 39 indoor sequences. This dataset allows the algorithm's performance to be evaluated under various complex conditions. In this paper, three sequences with a 640 × 480 resolution and 16-bit depth were chosen for the experiments: fr1-room, fr2-desk and fr3-long-office-household. Figure 10a–c shows sample images from the three sequences.

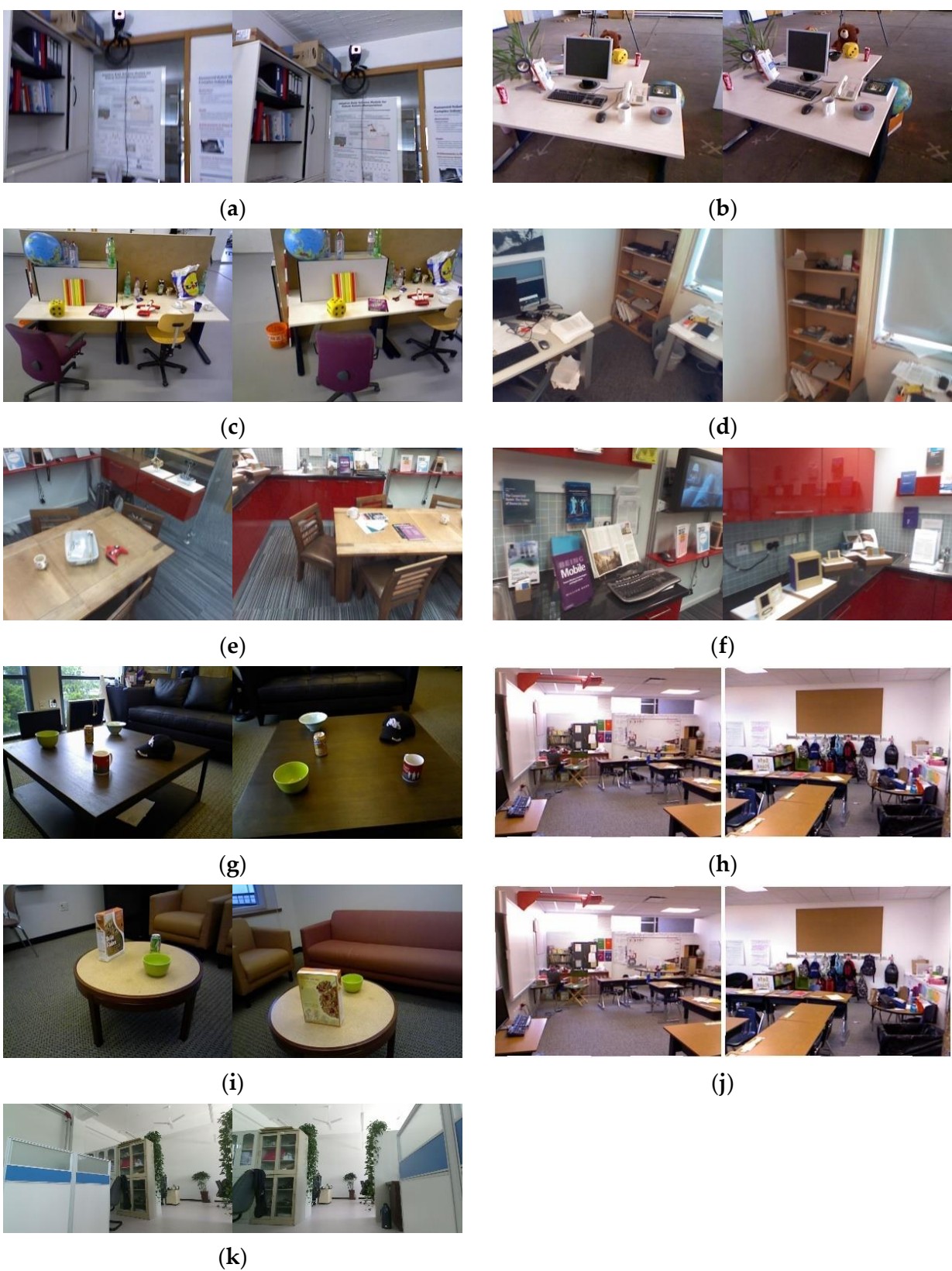

**Figure 10.** Sample images of datasets. (**a**–**c**) are the sample images of the TUM RGB-D dataset, (**d**–**f**) are the sample images of the Microsoft 7-Scenes dataset, (**g**–**i**) are the sample images of the RGBD Scenes dataset v2, (**j**) is the sample images of the NYUv2 dataset, and (**k**) is the sample images of the self-filmed dataset.

The Microsoft 7-Scenes dataset [35] is a public dataset filmed with a handheld Kinect camera that includes seven different indoor scenes, such as Chess, Office, RedKitchen, and Stairs, among others, and each scene is shot with multiple image sequences. Each sequence contains approximately 1000 RGB-D image frames. One image sequence from the Office scene and two image sequences from the RedKitchen scene were chosen for the loop closure detection experiments in this paper. The image has a 640 × 480 resolution and 16-bit depth. Figure 10d–f shows sample images of three sequences.

The RGBD Scenes dataset v2 [36] is a public dataset captured using the Microsoft Kinect sensor that is entirely oriented to indoor scenes and contains 14 different sequences of scene images, with furniture (chairs, coffee tables, sofas and tables) and objects (bowls, hats, cereal boxes, coffee cups, and cans) present in the scene images. The image content is relatively simple and can be used for image recognition, target detection, etc. In this paper, three image sequences (seq-03, seq-06 and seq-09) were chosen for the experiments, with a resolution of 640 × 480 and a 16-bit depth. The sample images of the two sequences are shown in Figure 10g–i.

The NYUv2 dataset [37] is made up of image sequences of various indoor scenes captured by the Microsoft Kinect sensor, and it contains 1449 annotated scene images and 407,024 unannotated images, as well as 894 semantic labels. Due to the excessive number of semantic labels in the dataset, we used the method proposed in [38] to divide the objects into 40 semantic labels, and the Deeplabv3+ model was trained and tested using the 1449 labeled images. This dataset's sample images are shown in Figure 10j.

The self-filmed dataset is a dataset independently filmed by our research group in the laboratory environment, which contains the images under light changes, object changes and other situations. The goal is to test the loop closure detection algorithm's adaptability to various environmental changes. The resolution of the collected images is 960 × 540, and the depth is 16 bits. The images were taken in an 8 m × 9 m laboratory environment including tables, chairs, bookshelves, people and other objects, which is a relatively complex indoor environment. Figure 10k represents a sample image from this dataset.

### 4.2. Pretraining of Semantic Segmentation Model

The Deeplabv3+ semantic segmentation model must first be pretrained before conducting formal experiments. The original Deeplabv3+ semantic segmentation model takes too long to train and run, and it cannot meet the real-time requirements of loop closure detection because it uses Xception as the backbone network. In this paper, the Deeplabv3+ segmentation model's backbone networks are Xception, Resnet50 and MobileNetv2, and the final applicable backbone networks are determined with the test results. Accuracy (ACC), mean intersection over union (mIoU), training time per epoch, time to complete the entire model training and the trained model's time to process each image are the evaluation metrics. Table 1 shows the running hardware and software configurations, as well as the program compilation language, of Deeplabv3+ semantic segmentation models equipped with various backbone networks, and Table 2 shows the test results.

**Table 1.** Configuration of training environment for semantic segmentation model.

| Hardware Configuration | | Software Configuration | |
|---|---|---|---|
| Operating system | Windows 10 | GPU-Driver | 457 |
| CPU | Intel i7 10700 | CUDA | 11.1 |
| Internal storage | 16 GB | Python | 3.8 |
| Hard disk | SSD 2TB | Torch | 1.8.1 |
| Graphics card | GEFORCE RTX 3070 | | |

**Table 2.** Test results of Deeplabv3+ model with different backbone networks.

| Backbone Network | ACC | mIoU | Training Duration of Each Epoch | Training Duration of The Model | Processing Duration of Each Image |
|---|---|---|---|---|---|
| Xception | 75.68% | 74.11% | 3.18 min | 10.6 h | 0.141 s |
| Resnet | 75.41% | 73.62% | 4.02 min | 13.4 h | 0.125 s |
| MobileNetv2 | 73.97% | 71.86% | 1.17 min | 3.9 h | 0.078 s |

The Deeplabv3+ semantic segmentation model, equipped with various backbone networks, is pretrained and tested in this paper using the public RGB-D dataset PASCAL VOC 2012. Each backbone network is trained for 200 epochs, and the total training time is the sum of those times.

Table 2 shows that MobileNetv2 takes significantly less time to train and run than Xception and Resnet, making it more suitable for real-time loop closure detection, with little difference in the accuracy (ACC) and mean intersection over union (mIoU) results. Furthermore, MobileNetv2 requires less memory space and computing power than Xception and Resnet. As a result, MobileNetv2 is chosen as the Deeplabv3+ model's backbone network, and the Deeplabv3+ semantic segmentation model is optimized to reduce the algorithm's overall running time.

Following the establishment of the piggyback backbone network, the optimized Deeplabv3+ network is trained on the NYUv2 dataset. After training the optimized Deeplabv3+ semantic segmentation model to achieve the desired segmentation effect, we will use it to semantically segment the scene images and complete the loop closure detection experiments using the semantic segmentation results.

*4.3. Ablation Experiment*

This section describes the experiments used to test and compare the performance of each component of the algorithm and the overall algorithm in this paper. The ablation experiments are divided into three groups in order to test the three algorithm components and the overall algorithm.

1.  Ours-sem: In this paper's algorithm, only semantic nodes are constructed, and the semantic similarity score is used as the local similarity score in the calculation of the final similarity score in order to determine whether there is a loop closure between the current frame and the loop closure candidate frame, but the appearance features described in Section 3.2.3 are not used in this experiment;

2.  Ours-app: Only appearance features are used for comparison, and the appearance similarity score is used instead of the local similarity score in the calculation of the final similarity score, which relies solely on the appearance features in the current and candidate frames to complete the loop closure detection without constructing and comparing the semantic nodes described in Section 3.2.2;

3.  Ours: This is the complete algorithm proposed in this paper.

In this section of the experiment, 50 image sequences were chosen for algorithm performance testing from a pool of four datasets containing a total of 80 loop closures. The three algorithms listed above are required to accurately determine the number of loop closures in the image sequences. The number of loop closures correctly judged by each algorithm and the accuracy rate were counted after the experiment to demonstrate the performance of the algorithms, and the test results are shown in Table 3.

**Table 3.** Loop closure judgment results of each component of our algorithm.

|  | Ours-Sem | Ours-App | Ours |
|---|---|---|---|
| Correctly judged loop closures | 54 | 36 | 70 |
| Precision rate | 67.5% | 45% | 87.5% |

The experimental data in Table 3 clearly show that the complete algorithm proposed in this paper outperforms Ours-sem and Ours-app with 20% and 42.5% higher accuracy rates, respectively. It is difficult to accurately determine the loop closure contained in the image sequence using only semantic and appearance features, and using only semantic features produces significantly better results than using only appearance features. This demonstrates that first constructing semantic nodes for judging image similarity through semantic features is reasonable, and that constructing similar node pairs by comparing semantic nodes leads to strong filtering, which can assist the algorithm in avoiding a large number of redundant calculations and improves the performance of the algorithm.

Furthermore, the proposed algorithm includes three filtering mechanisms: the screening of loop closure candidate frames via Gist similarity, the screening of candidate frames with a higher similarity to the current frame via semantic label similarity and, finally, the screening of similar node pairs of semantic nodes via cosine similarity. These mechanisms are used to eliminate a large number of images with too little similarity to the current frame and improve the efficiency of loop closure detection. In this paper, the screening mechanism in the algorithm is separated and compared in terms of time to verify whether the proposed algorithm can effectively reduce the running time. Ablation experiments were carried out for the four algorithm components listed below, as well as for the overall algorithm.

1. Ours-gist: Only the screening mechanism of filtering loop closure candidate frames via the Gist similarity score is retained, and the rest of the feature extraction and similarity calculation methods are not changed;
2. Ours-label: Only the filtering mechanism of node pairs via semantic label similarity is retained, and the rest of the feature extraction and similarity calculation methods are not changed;
3. Ours-sim: Only the filtering mechanism of similar node pairs via cosine similarity is retained, and the rest of the feature extraction and similarity calculation methods are not changed;
4. Ours: This is the complete algorithm proposed in this paper.

Using the above four algorithms to detect 10 image sequences, with each sequence containing 200 images, the time spent by each algorithm for loop closure detection was counted, and the running time of the three algorithms was compared. The test results are shown in Table 4.

**Table 4.** Running time of each screening component.

|  | **Ours-Gist** | **Ours-Label** | **Ours-Sim** | **Ours** |
|---|---|---|---|---|
| 1 sequence | 12.7132 s | 18.4235 s | 18.6542 s | 11.8663 s |
| 2 sequence | 14.6954 s | 18.7271 s | 18.7868 s | 13.8740 s |
| 3 sequence | 11.1627 s | 18.3247 s | 18.5454 s | 10.4061 s |
| 4 sequence | 15.5036 s | 18.2233 s | 18.6237 s | 14.6953 s |
| 5 sequence | 14.5001 s | 17.9893 s | 18.6766 s | 13.6915 s |
| 6 sequence | 13.3439 s | 18.1316 s | 18.5647 s | 12.5051 s |
| 7 sequence | 13.4348 s | 18.5023 s | 18.8021 s | 12.5364 s |
| 8 sequence | 11.8122 s | 17.9884 s | 18.6628 s | 10.9572 s |
| 9 sequence | 15.2271 s | 18.3266 s | 18.5782 s | 14.4254 s |
| 10 sequence | 12.9023 s | 18.1192 s | 18.7065 s | 11.0443 s |
| Average | 13.5295 s | 18.2756 s | 18.6601 s | 12.6001 s |

The algorithm's average running time is 13.5295 s after the Gist filtering mechanism is introduced, and 18.2756 s and 18.6601 s after the semantic labeling and similarity pair filtering mechanisms are introduced, respectively. This is due to the Gist filtering mechanism's ability to significantly reduce the number of images processed by the subsequent semantic segmentation model, effectively shortening the loop closure detection algorithm's running time. In contrast, using only semantic labels or similar node pairs for screening has no effect on the algorithm's running time. By incorporating the three screening mechanisms mentioned above into the proposed algorithm, the running time of the loop

closure detection algorithm is significantly reduced, which can better meet the real-time requirements of VSLAM and is more conducive to the algorithm's implementation and application in the real world.

*4.4. Loop Closure Judgment*

In this part of the experiment, we used three sequences of fr1-room, fr2-desk and fr3-long-office-household from the TUM RGB-D dataset, three sequences from the Microsoft 7-Scenes dataset, three sequences from the RGBD Scenes dataset v2 and one sequence from the self-filmed dataset, totaling 10 image sequences, to validate the proposed algorithm for loop closure detection.

In reality, the images taken frequently have illumination change, viewpoint deviations, and missing or changing objects. We classified sequences based on these three cases and selected three sets of image sequences from four datasets, each of which contained cases of illumination change, viewpoint deviation and object change. As shown in Figure 11, 15 subsequences (a total of 60 subsequences) were extracted from each dataset to form the experimental image set for that case. There may or may not be loop closures among all the subsequences in the experiment, which must be judged by the algorithm itself; therefore, the number and accuracy of the correct ones were statistically calculated at the end of the experiment. Additionally, in order to verify the performance of the algorithms in terms of time, the average processing time of each loop closure detection for a single image in different subsequences also needs to be calculated.

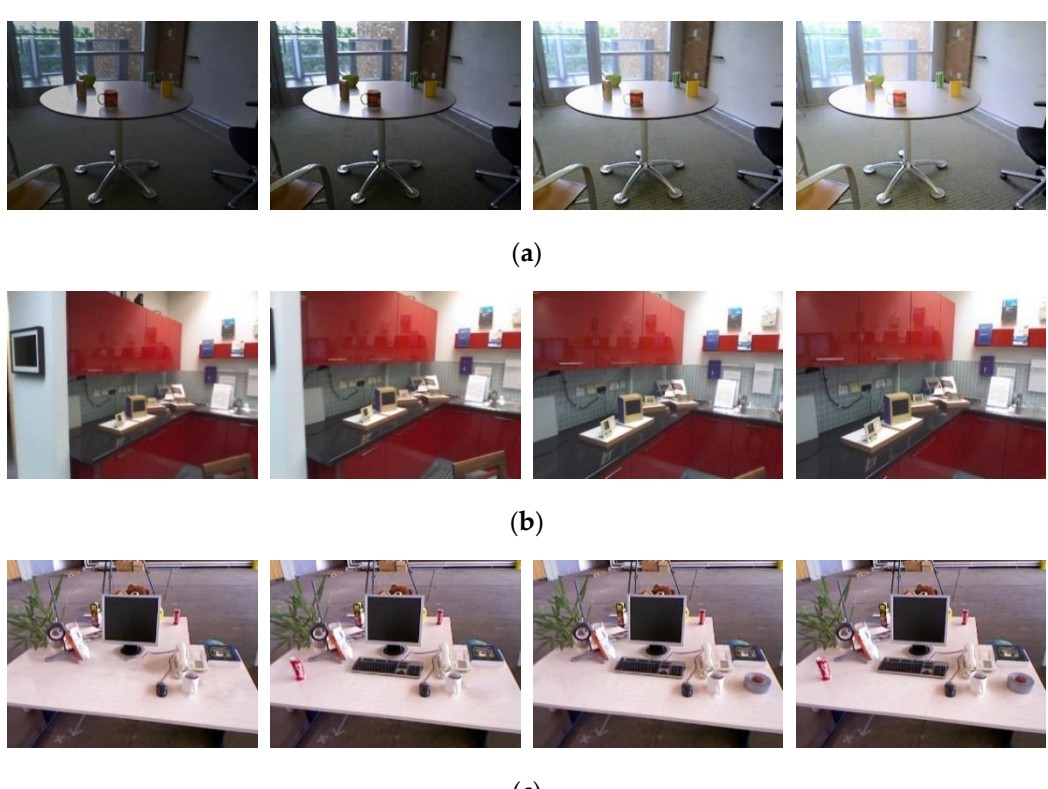

(**a**)

(**b**)

(**c**)

**Figure 11.** Examples of three sets of experimental image sequences: (**a**) a subsequence with illumination change between images, (**b**) a subsequence with mild viewpoint shifts between images, and (**c**) a subsequence with objects moving and missing between images.

To analyze a large amount of experimental data, the thresholds involved in Section 3 were set as follows: in the process of obtaining the Gist similarity score, the Gist similarity score threshold $\theta_g$ was set to 0.9; in the process of removing noise, the noise mask-block area-proportion threshold $\theta_a$ was set to 0.01; and in the process of calculating the semantic similarity score, the semantic label similarity number threshold $\theta_l$ was set to 0.75, the cosine

similarity threshold of node pairs $\theta_c$ was set to 0.99 and the similar node pairs percentage threshold $\theta_s$ was set to 0.75. Three sets of image sequences were tested using our proposed algorithms, the DBoW, CNN-W, Cao. and Wu., and Tables 5–7 show the results. Among them, Table 5 shows the test results in the presence of illumination changes, Table 6 shows the test results in the presence of viewpoint deviations and Table 7 shows the test results in the presence of moving and missing objects.

**Table 5.** Comparison results in the case of illumination changes.

|  |  | Ours | DBoW | CNN-W | Cao. | Wu. |
|---|---|---|---|---|---|---|
| TUM RGB-D dataset | The number of correct judgments | 12 | 9 | 10 | 12 | 11 |
|  | Precision rate | 80.00% | 60.00% | 66.67% | 80.00% | 73.33% |
|  | Average processing time per image | 66.8 ms | 3.2 ms | 122.3 ms | 168.4 ms | 482.2 ms |
| Microsoft 7-Scenes dataset | The number of correct judgments | 13 | 9 | 11 | 12 | 12 |
|  | Precision rate | 86.77% | 60.00% | 73.33% | 80.00% | 80.00% |
|  | Average processing time per image | 64.2 ms | 2.9 ms | 120.8 ms | 167.4 ms | 485.2 ms |
| RGBD Scenes dataset v2 | The number of correct judgments | 14 | 10 | 12 | 12 | 13 |
|  | Precision rate | 93.33% | 66.67% | 80.00% | 80.00% | 86.77% |
|  | Average processing time per image | 65.1 ms | 3.0 ms | 122.2 ms | 166.8 ms | 483.7 ms |
| Self-filmed dataset | The number of correct judgments | 13 | 10 | 11 | 12 | 12 |
|  | Precision rate | 86.77% | 66.67% | 73.33% | 80.00% | 80.00% |
|  | Average processing time per image | 64.1 ms | 2.9 ms | 121.9 ms | 168.6 ms | 485.0 ms |
| All | The number of correct judgments | 52 | 38 | 44 | 48 | 48 |
|  | Precision rate | 86.67% | 63.33% | 73.33% | 80.00% | 80.00% |
|  | Average processing time per image | 65.1 ms | 3.0 ms | 121.8 ms | 167.8 ms | 484.0 ms |

**Table 6.** Comparison results in the case of viewpoint deviations.

|  |  | Ours | DBoW | CNN-W | Cao. | Wu. |
|---|---|---|---|---|---|---|
| TUM RGB-D dataset | The number of correct judgments | 13 | 9 | 11 | 12 | 12 |
|  | Precision rate | 86.77% | 60.00% | 73.33% | 80.00% | 80.00% |
|  | Average processing time per image | 67.5 ms | 3.0 ms | 125.1 ms | 170.0 ms | 489.8 ms |
| Microsoft 7-Scenes dataset | The number of correct judgments | 14 | 9 | 12 | 12 | 13 |
|  | Precision rate | 93.33% | 60.00% | 80.00% | 80.00% | 86.77% |
|  | Average processing time per image | 66.1 ms | 2.9 ms | 122.2 ms | 168.8 ms | 487.1 ms |
| RGBD Scenes dataset v2 | The number of correct judgments | 14 | 10 | 12 | 13 | 13 |
|  | Precision rate | 93.33% | 66.77% | 80.00% | 86.77% | 86.77% |
|  | Average processing time per image | 69.4 ms | 2.8 ms | 122.7 ms | 167.2 ms | 486.4 ms |
| Self-filmed dataset | The number of correct judgments | 13 | 9 | 12 | 13 | 13 |
|  | Precision rate | 86.77% | 60.00% | 80.00% | 86.77% | 86.77% |
|  | Average processing time per image | 70.3 ms | 3.1 ms | 123.6 ms | 169.5 ms | 488.3 ms |
| All | The number of correct judgments | 54 | 37 | 47 | 50 | 51 |
|  | Precision rate | 90.00% | 61.67% | 78.33% | 83.33% | 85.00% |
|  | Average processing time per image | 68.3 ms | 3.0 ms | 123.4 ms | 168.9 ms | 487.9 ms |

**Table 7.** Comparison results in the case of moving and missing objects.

|  |  | Ours | DBoW | CNN-W | Cao. | Wu. |
|---|---|---|---|---|---|---|
| TUM RGB-D dataset | The number of correct judgments | 13 | 10 | 12 | 12 | 13 |
|  | Precision rate | 86.77% | 66.77% | 80.00% | 80.00% | 86.77% |
|  | Average processing time per image | 69.6 ms | 3.1 ms | 124.4 ms | 170.1 ms | 488.6 ms |
| Microsoft 7-Scenes dataset | The number of correct judgments | 14 | 9 | 13 | 13 | 13 |
|  | Precision rate | 93.33% | 60.00% | 86.77% | 86.77% | 86.77% |
|  | Average processing time per image | 69.3 ms | 3.0 ms | 124.0 ms | 169.0 ms | 487.8 ms |
| RGBD Scenes dataset v2 | The number of correct judgments | 14 | 10 | 13 | 14 | 14 |
|  | Precision rate | 93.33% | 66.77% | 86.77% | 93.33% | 93.33% |
|  | Average processing time per image | 68.5 ms | 2.9 ms | 123.7 ms | 168.5 ms | 487.5 ms |
| Self-filmed dataset | The number of correct judgments | 14 | 9 | 12 | 12 | 13 |
|  | Precision rate | 93.33% | 60.00% | 80.00% | 80.00% | 86.77% |
|  | Average processing time per image | 68.9 ms | 3.1 ms | 126.3 ms | 170.4 ms | 488.4 ms |
| All | The number of correct judgments | 55 | 38 | 50 | 51 | 53 |
|  | Precision rate | 91.67% | 63.33% | 83.33% | 85.00% | 88.33% |
|  | Average processing time per image | 69.1 ms | 3.0 ms | 124.6 ms | 169.5 ms | 488.1 ms |

According to the experimental data in Tables 5–7, it can be seen that the accuracy of the algorithm in this paper improves by 23.34% for loop closure detection compared to the DBoW, 13.34% compared to the CNN-W and 6.67% compared to Cao. and Wu. when there is an illumination change within the scene. With the presence of a viewpoint deviation in the image, the algorithm in this paper improves the accuracy by 28.33% compared to the DBoW, 11.67% compared to the CNN-W, 6.67% compared to Cao. and 5% compared to Wu. In the case of moving and missing objects in the image, the accuracy of the algorithm in this paper is improved by 28.34%, 8.34%, 6.67% and 3.34% compared with the four algorithms of DBoW, CNN-W, Cao. and Wu., respectively. We can see from the above experimental results that the DBoW performs poorly in all three cases, the CNN-W outperforms the DBoW, and the two advanced algorithms, Cao. and Wu., outperform the DBoW. However, when compared to the other four algorithms, the proposed algorithms' accuracy in loop closure detection in the same cases is improved, and the detection effect is better.

In addition, in terms of time, it is obvious that the average processing time of the loop closure detection algorithm based on deep learning for a single image is much longer. However, due to the screening mechanism of our proposed algorithm, a large number of redundant images are eliminated before being processed with the deep neural network, and thus, the computational cost of our proposed algorithm is significantly reduced. Compared with the DBoW, the computational cost of our algorithm increases, but the accuracy improves by more than 20%.

*4.5. Analysis of Precision-Recall*

The precision-recall (P-R) curve is one of the metrics used to verify the performance of the loop closure algorithm. The precision in the P-R curve refers to the proportion of real closed loops among the closed loops detected by the algorithm, that is, how many of the detected loop closures are truly loop closures. If the precision rate is too low, it means that a large number of loop closures are determined as false positives in the process of loop closure detection; the recall refers to the proportion of all true loop closures that are correctly detected, i.e., how many true loop closures can be correctly detected, and if the recall is too low, it means that most true loop closures cannot be detected.

In this part of the experiment, one image sequence from each of the four datasets mentioned in Section 4.1 was selected for testing: fr3-long-office-household from the TUM RGB-D dataset, one image sequence contained in the RedKitchen scene from the Microsoft 7-Scenes dataset, the seq-06 image sequence in RGBD Scenes dataset V2, and the image sequence in the self-filmed dataset. Experiments were conducted using keyframes in each image sequence, and multiple loop closures were artificially set in each sequence. The precision and recall of each loop closure detection algorithm were tested using the same image sequences each time the experiments were conducted, and the P-R curves were plotted. The four datasets' image sequence examples and loop closure examples are shown in Figure 12.

Each algorithm was tested using the four image sequences and the P-R curves were plotted based on the experimental result data; the results are shown in Figure 13. The green curve represents the proposed algorithm, the black curve represents the DBoW, the blue curve represents the CNN-W, the red curve represents the algorithm of Cao and the yellow curve represents the algorithm of Wu. Figure 13 shows that as the threshold value gradually falls, the number of loop closures detected by each algorithm gradually rises; then, the recall rises, but the precision rate gradually declines. In Figure 13, the performance of Wu's algorithm is better than that of the DBoW, CNN-W and Cao. in most cases, but in individual cases, such as in Figure 13b, when the recall is in the period from 0.49 to 0.68, the detection performance is lower than that of Cao., but is still higher than that of the CNN-W, whereas the DBoW algorithm performs the worst. Additionally, the detection performance of the proposed algorithm in this paper is generally higher than that of the other four algorithms. Under different datasets, the algorithm in this paper has high recall

and high precision, and the performance is better than that of the DBoW, CNN-W, Cao. and Wu. algorithms, which can effectively detect loop closures.

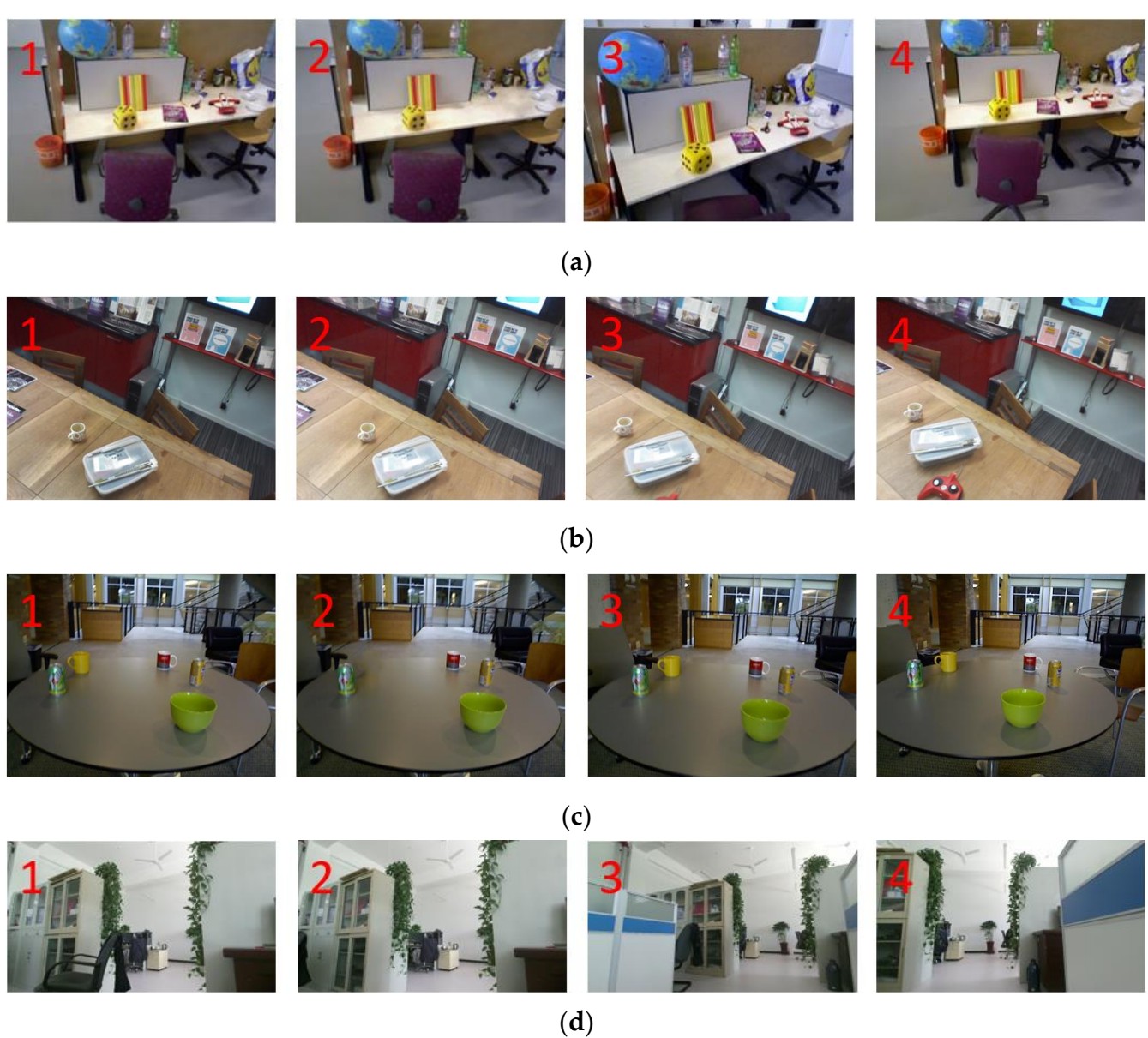

**Figure 12.** Four selected image sequence examples and loop closure examples in the precision-recall experiment. The two images with sequence number 1 and 2 are artificially set as true loop closures, and any other two images may not be true loop closures. (**a**) is the image sequence example of the TUM RGB-D dataset, (**b**) is the image sequence example of the Microsoft 7-Scenes dataset, (**c**) is the image sequence example of the RGBD Scenes dataset V2, and (**d**) is the image sequence example of the self-filmed dataset.

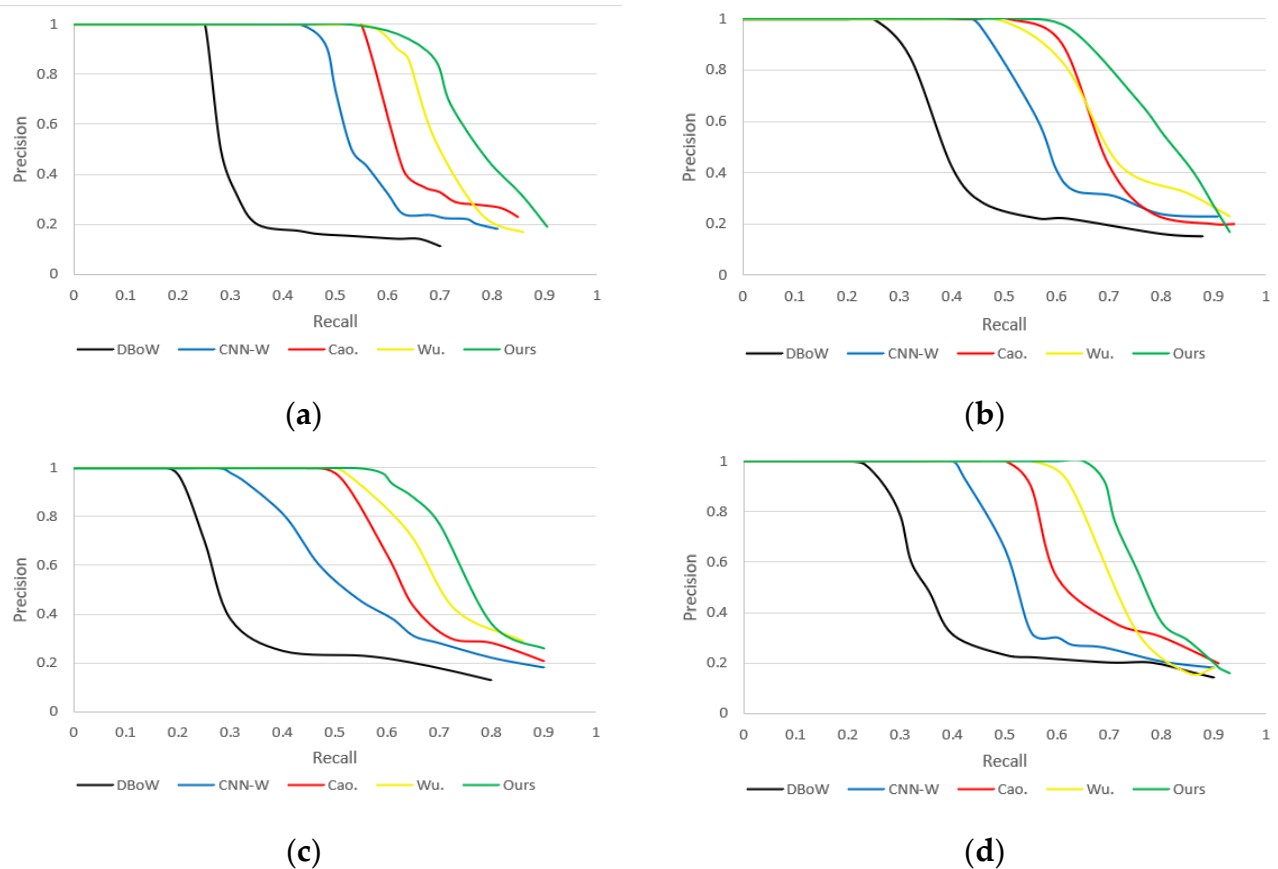

**Figure 13.** P-R curves: (**a**) shows the P-R curves corresponding to the fr3-long-office-household dataset, (**b**) shows the P-R curves corresponding to the RedKitchen dataset, (**c**) shows the P-R curves corresponding to the seq-06 dataset and (**d**) shows the P-R curves corresponding to the self-filmed dataset.

## 5. Conclusions

As an indispensable part of visual SLAM, loop closure detection can effectively reduce the impact of accumulated errors on positioning and mapping, and improve their accuracy. We propose a loop closure detection algorithm that incorporates Gist features, semantic features and appearance features in this paper. The optimized DeepLabv3+ network is used to semantically segment the images and exclude noise interference before the loop closure detection task is completed by screening for Gist similarity, semantic similarity and appearance similarity. The experimental results show that our proposed algorithm can efficiently and accurately detect loop closures in images of indoor environments. The algorithm, on the other hand, is more concerned with acquiring and utilizing local features. Our next major research topic is how to better balance global and local features, and thus, build a more reasonable image similarity comparison model to further improve image similarity comparison accuracy. The current semantic segmentation model, on the other hand, is limited by the semantic label classification of the training dataset and cannot segment objects other than labels, which is also a problem that needs to be addressed in our future work.

**Author Contributions:** Conceptualization, J.L., P.W. and C.N.; Methodology, J.L., C.N. and W.H.; Software, J.L. and W.H.; Validation, J.L. and D.Z.; Formal analysis, J.L., P.W., C.N. and W.H.; Investigation, J.L., P.W. and D.Z.; Resources, J.L. and P.W.; Data curation, J.L., C.N. and D.Z.; Writing—original draft, J.L.; Writing—review & editing, J.L., P.W., C.N., D.Z. and W.H.; Visualization, J.L.; Supervision, P.W. and C.N.; Project administration, P.W. and C.N.; Funding acquisition, P.W. All authors have read and agreed to the published version of the manuscript.

**Funding:** This research was funded by the China Postdoctoral Science Foundation (Grant No. 2021M702030) and the Science and Technology Project of Shandong Provincial Department of Transportation (Grant No. 2021B120).

**Institutional Review Board Statement:** Not applicable.

**Informed Consent Statement:** Not applicable.

**Data Availability Statement:** Not applicable.

**Conflicts of Interest:** The authors declare no conflict of interest.

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
