# Peer review of "Loop Closure Detection for Mobile Robot based on Multidimensional Image Feature Fusion"

_machines, doi:10.3390/machines11010016_

Round 1

Reviewer 1 Report

The authors propose a system for detecting loop closures in visual SLAM using three different image representation techniques. At first, Gist extracts the global feature; next, local ones are extracted through Neural Networks and moments. The pipeline is tested and validated in different public and self-generated datasets.

The article is well-organized, without grammatical issues in general. The related section is not proper for a journal manuscript. The description of the feature's extraction technique methodology is not in the scope of such a paper. The authors should mention works based on global features, whether Gist or HoG. Regarding the methodology, even if the authors describe their method clearly, I doubt its novelty. The reason behind this is that a lot of image description techniques exist, which makes the pipeline computationally costly. Moreover,  the definition of keyframes is missing. How do the authors select the keyframes in the incoming image stream? The following experiments, which are held only in indoor environments, do not present computational evaluation, while comparisons are performed with methods that are not new, e.g., DBoW. Many recent systems are open-source and can be used, but this does not constitute the main drawback of the article. 

One minor comment regards the introduction, where the loop closure definition should point to [1].

[1] K. A. Tsintotas, L. Bampis and A. Gasteratos, "The Revisiting Problem in Simultaneous Localization and Mapping: A Survey on Visual Loop Closure Detection," in IEEE Transactions on Intelligent Transportation Systems, vol. 23, no. 11, pp. 19929-19953, Nov. 2022, doi: 10.1109/TITS.2022.3175656.

Author Response

Thank you very much for your approval and comments. We have revised the manuscript as follows:

  1. Gist features are used in the manuscript to quickly screen the candidate frames, and the Section 2.1 has given a detailed description of Gist features. In the revised manuscript, we added the description and analysis of the loop closure detection method based on Gist in the second paragraph of "Introduction".
  2. Compared with traditional methods, the loop closure detection methods based on deep learning do consume more time and computational cost. We have taken this problem into consideration and added the steps of candidate frames screening using Gist features before semantic segmentation, so as to eliminate a large number of redundant images to be processed by deep neural network. In Table 5-7 of Section 4.4 of the revised manuscript, we have added comparative experimental data in terms of time. It can be seen that compared with other algorithms based on deep learning, the computational cost of our proposed algorithm is significantly reduced. Compared with DBoW, the computational cost of our algorithm increases, but the accuracy improves by more than 20%.
  3. In Section 3.1, we have added an explanation of keyframes and a method for extracting them.
  4. Among the comparison algorithms we selected, DBoW is a classic traditional loop closure detection algorithm, so we chose to compare with this algorithm to verify the gap between our proposed algorithm and the traditional algorithm. In addition, CNN-W is an algorithm proposed in 2019, and the algorithms proposed by Cao. and Wu. are new algorithms proposed in 2021.
  5. According to your suggestion, "The Revisiting Problem in Simultaneous Localization and Mapping: A Survey on Visual Loop Closure Detection" has been added to the revised manuscript as reference [2].

Reviewer 2 Report

-    This paper proposed a loop closure detection algorithm that fuses human-designed features, which is a GIST feature, and deep learning-based features, which are semantic and appearance features for high performance and low computational time.

-    The third paragraph of the introduction needs to be summarized. There is too much information of the previous work. Summarize them more succinctly or write them in the related work section.

-    In section 4.3, Table 4 should include the running time result of other loop closure detection algorithms used for comparison in Table 5. As Table 4 only contains the running time of the proposed method, it is hard to evaluate the contribution of using candidate frames to shorten the computational time.

-    The author should write the meaning before using abbreviations.

-    The manuscript needs to be professionally edited by a native speaker.

-    There are some minor comments

-       In pages 6 and 7, equations 2 and 3 have strangely expressed variables, such as m00. Does it mean m0 × 0 or m00 ?

-       In line no. 256, the same sentence is repeated twice.

-       In line no. 287, the sentence should start with capital letters

-       In figure 11, caption (a) is bolded, whereas captions (b) and (c) are not.

Author Response

Thank you very much for your advice. According to your suggestions, we have made the following modifications to the manuscript:

  1. We summarize and simplify the third paragraph of “Introduction”.
  2. In Table 5-7 of Section 4.4 of the revised manuscript, we have added comparative experimental data in terms of time. It can be seen that compared with other algorithms based on deep learning, the computational cost of our proposed algorithm is significantly reduced. Compared with DBoW, the computational cost of our algorithm increases, but the accuracy improves by more than 20%.
  3. We added the full name before the abbreviation.
  4. We have checked and proofread the manuscript again, and corrected grammatical errors and spelling mistakes.
  5. According to the comments you put forward, we have revised the details and checked the full paper.

Round 2

Reviewer 1 Report

The manuscript is ready for publication. My requirements are fulfilled. 

Author Response

Thank you very much for your approval.

Reviewer 2 Report

Overall: Most questions are answered adequately. A remaining comment is given below.

Previous review comment 4:

The manuscript needs to be professionally edited by a native speaker.

Your answer:

We have checked and proofread the manuscript again, and corrected grammatical errors and spelling mistakes.

Review comments:

The manuscript still needs to be professionally edited by a native speaker. Use a pay service. The manuscript has many grammatical errors, and many sentences are too long to understand the content clearly. For example:

-          Line no. 55: “et al” should be written as “et al.

-          Line no. 36-40
“The loop closure detection in the existing VSLAM system primarily determines whether the loop closure can be formed by calculating the similarity between the current frame and the keyframe, so loop closure detection is fundamentally a scene recognition and image matching problem [3,4], which compares the current scene in which the mobile robot is located with the historical scene to determine whether the scene is the same.”

-          Line no. 123-129
“Following that, the semantic labels output the segmentation results one by one, after obtaining the center of mass for each mask block in the segmentation results, the three-dimensional (3-D) semantic nodes can be built by combining the image depth information, and the cosine similarity of the semantic nodes in the two frames can be compared one by one to obtain the semantic similarity score of each node pair, the node pair with higher similarity is chosen as the similar node pair.”

-          These long sentences are difficult to comprehend. There are many more like these.

Previous review comment 5:

There are some minor comments

-          In pages 6 and 7, equations 2 and 3 have strangely expressed variables, such as m00. Does it mean m0 × 0 or m00 ?

-          In line no. 256, the same sentence is repeated twice.

-          In line no. 287, the sentence should start with capital letters

In figure 11, caption (a) is bolded, whereas captions (b) and (c) are not.

Your answer:

According to the comments you put forward, we have revised the details and checked the full paper.

Review comments:

In line no. 246, the same sentence is repeated twice

-          “but it also ensures the algorithm's accuracy. It also ensures the algorithm's accuracy."

Author Response

Thank you very much for your comments. We asked the English editing service of MDPI to help revise the manuscript, and then we proofread it again.

English Editing Certificate can be found in the attachment.
